# Is Olfactory Testing a Useful Diagnostic Tool to Identify SARS-CoV-2 Infections Early? A Cross-Sectional and Longitudinal Analysis

**DOI:** 10.3390/jcm12093162

**Published:** 2023-04-27

**Authors:** Christiana Graf, Inken Wagener, Katharina Grikscheit, Sebastian Hoehl, Annemarie Berger, Nils Wetzstein, Julia Dietz, Georg Dultz, Florian Michael, Natalie Filmann, Eva Herrmann, Peter Tinnemann, Udo Goetsch, Sandra Ciesek

**Affiliations:** 1Institute of Medical Virology, University Hospital Frankfurt, 60306 Frankfurt am Main, Germany; 2Department of Internal Medicine I, University Hospital Frankfurt, 60306 Frankfurt am Main, Germany; 3Department of Internal Medicine II, Infectious Diseases, University Hospital Frankfurt, 60306 Frankfurt am Main, Germany; 4Institute of Biostatistics and Mathematical Modeling, Goethe University, 60323 Frankfurt am Main, Germany; 5Public Health Department of the City of Frankfurt am Main, 60306 Frankfurt am Main, Germany; 6German Centre for Infection Research, Deutsches Zentrum für Infektionsforschung, External Partner Site Frankfurt, 60306 Frankfurt am Main, Germany

**Keywords:** SARS-CoV-2 infection, COVID-19 infections, olfactory dysfunction, olfactory test

## Abstract

BACKGROUND: Genesis and the prognostic value of olfactory dysfunction (OD) in COVID-19 remain partially described. The objective of our study was to characterize OD during SARS-CoV-2 infection and to examine whether testing of OD may be a useful tool in clinical practice in order to early identify patients with SARS-CoV-2 infection. METHODS: Olfactory function assessment was objectively carried out using the u-Smell-it^®^ test. In a cross-sectional study part, we evaluated this test in a control cohort of SARS-CoV-2 negative tested patients, who attended the University Hospital Frankfurt between May 2021 and March 2022. In a second longitudinal study part, sensitivity and specificity of OD was evaluated as a diagnostic marker of a SARS-CoV-2 infection in Frankfurt am Main, Germany in SARS-CoV-2 infected patients and their close contacts. RESULTS: Among 494 SARS-CoV-2 negative tested patients, OD was detected in 45.7% and was found to be significantly associated with the male gender (*p* < 0.001), higher age (*p* < 0.001), cardiovascular and pulmonary comorbidities (*p* < 0.001; *p* = 0.03). Among 90 COVID-19 positive patients, OD was found in 65.6% and was significantly associated with male gender and positive smoking status (*p* = 0.04 each). Prevalence and severity of OD were significantly increased in infections with the Delta variant (B.1.617.2) compared to those with the Omicron variant (BA.1.1.529). Diagnostic sensitivity and specificity of OD for diagnosis of SARS-CoV-2 infection were 69% and 64%, respectively. CONCLUSION: OD is common in COVID-19 negative and positive tested patients with significantly different prevalence rates observed between different variants. Diagnostic accuracy of OD is not high enough to implement olfactory testing as a tool in diagnostic routine to early identify patients with a SARS-CoV-2 infection.

## 1. Introduction

The global outbreak of coronavirus disease 2019 (COVID-19) induced by severe acute respiratory syndrome coronavirus (SARS-CoV-2) has emerged as one of the most significant global public health concerns in the twenty-first century and it was announced as a pandemic by World Health Organization (WHO) on 11 March 2020. While it was initially supposed that SARS-CoV-2 can only be transmitted by close contacts through naso/oropharyngeal droplets, soon evidence emerged that airborne transmission via inhalation of viral aerosols and fecal-oral transmission by COVID-19 positive stool or water sources represent a major transmission route for COVID-19 [1]. The rapid spread of COVID-19 has prompted robust public health investigations to characterize the disease and to analyze the virus in order to find suitable prevention and treatment strategies. Early published reports have reported a wide range of symptoms which included fever, cough, shortness of breath, fatigue, muscle or body aches, headache, sore throat, nausea or vomiting, diarrhea and new loss of taste or smell as key symptoms of COVID-19, which have been predominantly used for case identification and testing prioritization [2,3,4,5]. Chemosensory deficits have been observed to be transient during the course of COVID-19 infection, with patients regaining their smell and taste after several days to weeks.

The possible main route of transmission is thought to be the close contact and respiratory droplets. Therefore, it is necessary to maintain physical distance and wear a face mask. Other routes of transmission are through contaminated surfaces, as well as airborne and fecal-oral transmission.

Principal causes of OD include viral infections derived by corona or influenza viruses. The causal factor is assumed to be nasal congestion as well as damage to the olfactory receptor neurons [6,7]. Further causes of olfactory impairment concern trauma and rhinosinusitis or nasal polyposis. Posttraumatic olfactory dysfunction may be due to rupture of the olfactory filaments or cerebral contusion. The causes of sinonasal olfactory dysfunction include inflammatory or mechanical factors, such as diversion of access to the olfactory epithelium. In recent years extensive investigations have also confirmed the association between neurodegenerative disease and olfactory impairment and demonstrated the high diagnostic significance of this link [8,9,10]. OD thus serves an early warning of the most frequent neurodegenerative diseases such as idiopathic Parkinson disease or Alzheimer dementia. Beyond that, olfactory loss has been observed as a result of toxic exposure. Accordingly, changes in smell, as well as taste, have been reported following Australian elapid bite, mainly for black snakes due to neurotoxic mediated olfactory bulb atrophy [11,12].

However, it was assumed that olfactory dysfunction (OD) driven by the coronavirus SARS-CoV-2 differs from other virus-associated deficits in its sudden onset and its rapid recovery [13,14,15,16,17]. Underlying pathophysiological mechanisms of how the virus affects the sense of smell and taste have been widely discussed. One potential mechanism for COVID-19-induced OD may be the fact that COVID-19 specifically target cells in the ear-nose-throat (ENT) tract, including the olfactory epithelium [18,19]. Further hypotheses that are under debate concern viral neuroinvasion, immune reaction and the involvement of local inflammatory mediators [20,21]. Neurological hypotheses assume that OD derives from SARS-CoV-2 induced injury of the olfactory bulb or other central brain structures or circuits. Clinical studies reflect this theory by a significant association between OD and neurological symptoms, especially headaches [22].

The primary diagnostic goal of any pandemic is to detect infections early in order to provide containment and reduce the number of new infections. Particularly at the beginning of the SARS-CoV-2 pandemic, there was a prolonged shortage of antigen and reverse transcription polymerase chain reaction (RT-PCR)-based diagnostic tests of naso- and oropharyngeal samples to verify SARS-CoV-2 infection [23]. Another problem, which still exists now, was the moderate sensitivity of many rapid antigen tests at the beginning of a SARS-CoV-2 infection, which often turn out to be positive very late during the course of a SARS-CoV-2 infection [24,25].

Moreover, rapid antigen tests are most commonly performed on upper respiratory samples by nasal swabs whose sensitivity depend on how exactly wiping is carried out. Besides technique-dependent sensitivity, nasal testing has the disadvantage of discomfort, especially for small children. Beyond that, false positive test results range from 2% to 37%. A positive with regard to nasal samples is that nasal swabs are minimally invasive compared to other tests of saliva or other body fluids. Moreover, nasal testing allows social distancing: while some molecular tests, including RT-PCR, are sometimes conducted at a hospital or clinic, rapid antigen tests of nasal swabs can also be carried out at home. Moreover, fewer false negative test results are observed for nasal tests compared to others such as throat swabs or saliva tests.

One potential alternative to the preservation of respiratory and nasal probes for SARS-CoV-2 testing is the analysis of feces. Accordingly, previous analyses demonstrated that SARS-CoV-2 is detectable in feces of infected patients and is associated with gastrointestinal symptoms during a SARS-CoV-2 infection [26]. An advantage of testing fecal samples is the lack of invasiveness of sample collection, particularly in children. Furthermore, in contrast to oropharyngeal samples, the virus can be detected in stool probes over a long term period (up to 4 months) after infection occurred [26,27]. However, positive sampling rates of SARS-CoV-2 in feces have been observed to be relatively low with 29% compared to other probes such as oropharyngeal swabs (32–48%), sputum (72–76%) or nasopharyngeal swabs (63%) [28].

Nevertheless, feces testing has never gained acceptance in clinical routine due to the greater effort involved in sample collection and further processing of samples.

In the context of the controversy regarding sample collection, the lack of sensitivity and shortage of antigen tests, striking results have been published in recent meta-analyses, which assigned a major role to OD in early diagnosis and prognosis assessment of the disease: anosmia or hyposmia turned out to be not only the most frequent but also often the first symptom of a SARS-CoV-2 infection and, therefore, may be a helpful diagnostic tool in identifying the disease early [9,10]. An olfactory test in clinical practice for early detecting SARS-CoV-2 infections would be cheap, easy to practice and widely applicable. Olfactory tests most commonly used in clinical practice comprise the Snap & Sniff threshold test, which determines the odor detection threshold, the University of Pennsylvania Smell Identification Test (UPSIT) and the odor identification test with pen-like odor dispensing devices, which both are odor identification tests [29,30,31].

The aim of the following study was to investigate the usefulness of an olfactory test for early diagnosing and detecting SARS-CoV-2 infections. We therefore first evaluated the u-Smell-it^®^ test in a SARS-CoV-2 negative tested cohort in order to determine the proportion of those, who exhibit OD without COVID-19 infection and for whom this test is thus not applicable for early detecting a SARS-CoV-2 infection. In a second study part, we used this test in known COVID-19 index cases and their close contacts in quarantine in order to evaluate whether its application could be implemented in diagnostic routine to identify patients with a SARS-CoV-2 infection early.

## 2. Materials and Methods

### 2.1. Patients

The cross-sectional study part, which evaluated preexisting OD in a control group of SARS-CoV-2 negative tested patients, was conducted in the University Hospital of Frankfurt, Germany. All patients aged 18 and over with a documented negative SARS-CoV-2 PCR within the last 24 h, who sought medical care at the University Hospital Frankfurt, Germany between May 2021 and March 2022, were considered for enrollment in the study.

The longitudinal study part, which investigated the clinical course of OD in SARS-CoV-2 infected patients, was conducted on an outpatient basis by a collaboration with the Frankfurt Health Department, Germany. Ιn accordance with recommendations on quarantine and isolation of the Robert Koch Institute (RKI), testing was conducted on day 0 and day 5 of the beginning of quarantine and, in cases of a persistent test positivity or persistent clinical symptoms, repeatedly during the course of a SARS-CoV2- infection.

During May 2021 and April 2022, laboratory-confirmed COVID-19 index cases ≤ 1 day from diagnosis and their close contacts aged 7–65 years, who were in quarantine and absolved regular SARS-CoV-2 tests, were included in the study. Enrollment and follow-up of patients was performed in two distinct time periods. The first patient cohort was recruited during the time period between 1 May 2021 and 30 September 2021, in which the delta variant (B.1.617.2) was observed to be the predominant one according to RKI data [32]. Enrollment and follow-up of the second patient cohort was conducted between 1 December 2021 and 31 April 2022, which was predominantly characterized by the omicron variant (BA.1.1.529) [33].

Details on assessing baseline, clinical and epidemiological information of all participants in both study parts are found in Appendix A. The study was performed in accordance with the ethical guidelines of the Declaration of Helsinki and approved by the ethics committee of the Frankfurt University Hospital (ethics committee reference number 2021-96). All patients gave their written consent before being enrolled in the study. In cases of including participants <18 years of age, written consent was obtained by their parents or legal guardians.

### 2.2. Olfactory Function Assessment

Olfactory function assessment was carried out using the u-Smell-it^®^ test, which was developed by researchers from the University of Colorado Boulder and was approved by the FDA [32]. It is based on an index card, which contains five different scratch and sniff boxes with five different scents and a QR-code, by which the card can be coupled with a smartphone app. The participants have to scratch each of the five boxes, then smell them and select one out five answers in the app. After repeating this step for each box, a final result score of smelling capability (0 = no correct result, 5 = five correct results) is presented on the smartphone app. Details on grading of OD based on these test results is found in Appendix A.

In the cross-sectional study part, participants conducted this smell test once at enrollment of the study. In the longitudinal study part, loss of smell was analyzed using this smell test in SARS-CoV-2 infected individuals and their close contacts at enrollment of the study and afterwards every 2–3 days. For all close contacts who did not test positive, smell testing was performed for 10 days until SARS-CoV-2 infection could be refuted with certainty. Close contacts who tested positive during quarantine performed smell tests over an observation period of 28 days after symptom onset.

### 2.3. Statistical Analysis

Statistical analyses were performed using the IBM SPSS 26.0 statistical software package (SPSS/IBM, Munich, Germany). Clinical characteristics of patients were expressed as median and range. Categorial variables were compared using the χ^2^ or the Fisher’s exact test. Continuous variables were compared using the Mann–Whitney *U* test. We calculated the positive, the negative predictive value (PPV) and referred 95% confidence interval (CI) of olfactory dysfunction for close contacts of index cases using crosstabs and the χ^2^ test. All tests were two-sided and a *p* value of less than 0.05 was judged to be statistically significant.

## 3. Results

### 3.1. Cross-Sectional Study Part

Data on sociodemographic and health related factors, as well as on the severity of OD, of our study subjects are presented in Table 1. We enrolled 524 participants in the cross-sectional part of the study, of whom 307 (61.1%) were male and 523 (99.8%) were adults (median, 63 years [range 16–94, years]). Pre-existing medical conditions were recorded in 477 patients (91.0%; Table 1). Smoking as a potential olfactory risk factor was observed in 87 (16.6%) of patients.

OD of participants of the cross-sectional study part is detailed in Table 1. OD was present in 225 (42.9%) subjects, whereas 299 (57.1%) were considered normosmic. OD was observed in form of hyposmia in 204 (38.9%) of patients, while 21 (4.0%) were functionally anosmic. Mild hyposmia (grade 1) was observed in 105 patients (20.0%), moderate to severe hyposmia was observed in 65 and 34 patients, respectively (12.4% and 6.5%).

Characterizing patients with an OD, loss of smell was significantly more frequently observed in male patients (*p* < 0.001) compared to women. Analyzing OD according to different age groups, we observed significantly higher rates of OD in patients aged >65 years (*p* < 0.001; Figure 1A). In contrast, OD was found significantly less frequently in patients aged 16–55 years (*p* = 0.04, *p* < 0.001, *p* < 0.001, *p* = 0.002 and *p* = 0.007, respectively).

Beyond that, we found significant associations between cardiovascular disease (CVD) and OD as well as between pulmonary diseases and OD (*p* < 0.001; *p* = 0.02, Table 2). In contrast, further comorbidities, as well as positive smoking status, did not impact olfactory abilities in SARS-CoV-2 negative tested patients (Table 2). In order to determine independent predictive factors of OD in COVID-19 negative tested patients, we conducted logistic regression analysis. At univariate analysis, age, gender, CVD and pulmonary diseases were significantly associated with OD (*p* = <0.001; *p* < 0.001; *p* < 0.001; *p* = 0.02; Appendix A). A following multivariable analysis revealed that age and gender had largest effect on OD and were the only independent predictive factors of OD (*p* < 0.001; *p* < 0.001).

### 3.2. Longitudinal Study Part

A total of 212 consecutive patients were enrolled in the study, of whom 24.0% were male and 90 were tested positive for SARS-CoV-2 a median of one day after enrolment in the study. Among 90 participants tested positive for SARS-CoV-2, 131 individuals (52.5%) were male and 72 (80%) were adults (median, 29 years [range 7–65]). The main baseline patient characteristics of patients who tested positive for SARS-CoV-2, including demographic and clinical features, are listed in Table 3. Among those individuals with laboratory-confirmed SARS-CoV-2, 68 (75.6%) reported clinical COVID-19 symptoms (Table 3). A total of 22 (24.4%) of those individuals with positive test results remained asymptomatic throughout the whole course of infection. Of the remaining SARS CoV-2 infected participants, 28.9% each reported moderate and severe symptoms, while 15.6% experienced a mild course of disease (Appendix A).

A total of 59 out of 90 patients (65.6%), that have been tested positive for SARS-CoV-2 during quarantine, presented with OD during the course of disease. The majority of participants developed mild or moderate OD (17.8% and 18.9%, Appendix A). Severe OD was observed in 10%, a complete anosmia in 18.9%. Regarding prevalence of OD among different age groups, we observed that patients aged 56–65 years significantly more frequently developed impairment of smell during the course of SARS-CoV-2 infection (*p* < 0.001; Figure 1B). However, all further age groups did not significantly differ regarding the frequency of OD during a SARS-CoV-2 infection (Figure 1B). Accordingly, we observed no significant association between age and OD (*p* = 0.08; Table 4). By analyzing OD according to further baseline and clinical characteristics, we observed that male gender and positive smoking history were the only factors in SARS-CoV-2 infected individuals being associated with a lack of smell (*p* = 0.04; *p* = 0.04; *p* = 0.04; Table 4). On the contrary, olfactory function was not observed to be affected by specific clinical symptoms of SARS-CoV-2 infection.

Median time between diagnosis of SARS-CoV-2 infection and onset of OD was 4 days (range, 0–18). In 5 of those patients (5.5%), OD was observed before a positive SARS-CoV-2 finding. The remaining cases developed OD after the diagnosis of SARS-CoV-2 had been established. Median duration of OD was 4 days (range, 0–12) and recovery was observed a median of 7 days (range, −2–19) after SARS-CoV-2 diagnosis had been established (Appendix A). At the end of the observation period (28 days after OD onset), 4 patients (4%) still suffered from OD.

Enrollment and follow-up of patients was performed in two distinct time periods. During the first period of time (1 May–30 September 2021) COVID-19 infection events in Germany were primarily characterized by the Delta variant according to the RKI [33]. Enrollment and follow-up of the second patient cohort (*n* = 48) was conducted between 1 December 2021 and 31 April 2022, a time period which was characterized by the Omicron variant. Therefore, sub-analyses were performed for both longitudinal sub-cohorts in order to characterize OD in the context of different SARS-CoV-2 variants.

Interestingly, we found a significantly higher prevalence and severity of OD in patients infected with the Delta variant compared to those who suffered from the Omicron variant (prevalence: 79% vs. 56%, *p* = 0.03; severity: OD grade 2.2 vs. OD grade 0.9, *p* < 0.001, respectively). Regarding the temporal course of OD, no relevant differences could be detected between both variants (median time point of onset: *p* = 0.54, median time point of recovery: *p* = 0.23; median duration of OD: *p* = 0.08). Accordingly, comparable results could be detected concerning analysis of OD according to different baseline and clinical factors: male gender and positive smoking history were significantly associated with a lack of smell in both sub-cohorts. Beyond that, no further correlation was found between baseline parameters, clinical factors and SARS-CoV-2-related hyposmia (Appendix A).

Finally, we analyzed the diagnostic validity of objectively tested OD for the diagnosis of a SARS-CoV-2 infection. The results showed a sensitivity and specificity of 69% (95% CI, 58–78) and 64% (CI 95%, 43–82) and a corresponding positive predictive value of 87% (CI 95%, 77–94).

## 4. Discussion

Understanding OD in the general population is highly relevant, particularly in the wake of the global COVID-19 pandemic, which has led to increases in clinic visits and olfactory testing due to concerns for COVID-19.

In order to characterize OD in a control group, we conducted a cross-sectional analysis of OD in SARS-CoV-2 negative tested patients seeking medical care at the University Hospital Frankfurt. Our results demonstrated that OD seems to be common in this cohort with a prevalence of 42.9%. However, it must be considered that our cohort was primarily characterized by elderly patients: a total of 21 patients in the cohort (41.1%) were 65 years or older, most of whom presented with a significantly more pronounced impairment of olfactory function than younger patients. The results of several previous epidemiological studies confirm this observation. Decreases in the turnover of interneurons in the olfactory bulb (OB) and reduced activity in the olfactory cortex under olfactory stimulation have been supposed to cause these age-related changes in human olfaction [34]. To limit this age-related bias, we conducted sub-analyses of OD prevalence according to different age groups (Figure 1A). According to previous observations, our results showed that prevalence of smell dysfunction was significantly lower in patients aged 64 years and younger [34].

Regarding influence of different baseline and clinical factors on OD, male gender was not only significantly associated with OD but also turned out to be an independent predictive factor of impairment of smell. Gender differences in human olfaction in the general population have been widely discussed in the literature, with most of the studies confirming our observation of a female superiority in olfactory perception [35,36]. Several data suggest that neuroendocrine, social and cognitive elements may play a major role in this context [37,38]. In this context, previous studies have shown that OD is one of the first symptoms in several neurodegenerative diseases, such as Parkinson’s disease and Alzheimer’s disease, and appears years before motor symptoms and cognitive decline clinically manifest, which demonstrated that cognitive pathways play a major role in the pathogenesis of OD [39].

Moreover, we found a significant association between OD and CVD as well as between OD and pulmonary diseases, consistent with previous data. Nevertheless, especially regarding the relationship between CVD and OD, controversy results exist. The inconsistency of results may derive from varying age distributions. Accordingly, patients with CVD and pulmonary diseases in our study cohort were significantly older than patients not suffering from these comorbidities (CVS vs. non-CVD: 71 vs. 54 years; *p* < 0.001; pulmonary diseases vs. non-pulmonary diseases: 71 vs. 63; *p* = 0.002). Moreover, multivariate regression analysis revealed that age but not CVD and pulmonary diseases turned out to be independent predictive factors of OD. Thus, age-related changes in human olfaction might bias observed associations between CVD, pulmonary diseases and OD. Nevertheless, due to the highly significant associations observed between CVD and OD as well as between pulmonary diseases and OD, it is reasonable to assume that there are specific pathophysiological processes connecting CVD, pulmonary diseases and OD.

In our longitudinal study part, OD was found in 65.6% of SARS-CoV-2-positive tested patients, which is in line with several previous data. However, large discrepancies have been observed concerning the prevalence of OD reported by different studies with rates ranging from 5% to 98%. One reason for these discrepancies may be the different measurement instruments used to identify and classify OD, which might be highly variable when compared to each other. Beyond that, limited replicability of these tests across different countries may also prevent direct comparison of different data on OD prevalence. The olfactory test we used was developed and approved in the USA and the odors used to quantify olfactory ability are common there as well as in Europe. Thus, we assume that the reliability of olfactory test results in our German population is high.

OD most often occurred within the first two weeks from time point of SARS-CoV-2 diagnosis. Complete recovery rate of OD was observed in 96% of patients within the first four weeks. Regarding onset of OD, several studies reported that OD precedes other clinical symptoms and may therefore be indicative of COVID-19 infection [40,41].

Nevertheless, in our study results, OD was observed to occur concurrently with further clinical symptoms and in only five cases OD appeared before SARS-CoV-2 was diagnosed. Similar findings have also been reported in previous studies, which demonstrates that the clinical course seems to vary maybe due to different COVID-19 variants and further investigation is needed to characterize OD in COVID-19 infections [42,43].

Our results of the relatively late occurrence of OD in SARS-CoV2 infection together with the high prevalence of preexisting OD in the general population demonstrate that impairment of olfactory function may help to indicate SARS-CoV-2 infections but does not seem to be a reliable and sensitive screening tool in order to early detect SARS-CoV-2 infections. The sensitivity and specificity of objectively evaluated OD in our cohort was 69% (95% CI, 58–78) and 64% (95% CI, 43–82), respectively. The corresponding positive predictive value was 87% (95% CI, 77–94). Thus, OD is certainly more sensitive and specific than many other clinical symptoms in predicting the diagnosis of SARS-CoV-2 infections. However, a sensitivity of 69% is not sufficient enough to implement the detection of OD in diagnostic routine to early identify a SARS-CoV-2 infection.

Besides the use of different measurement instruments, variable strains of SARS-CoV-2 in various countries, as well as varying pathogenicity for the nasal cavity by different strains, may also contribute to variable observations of smell dysfunction in SARS-CoV-2 infections. Accordingly, our sub-analysis results showed that prevalence as well as severity of OD differed significantly between both strains with higher rates and a higher severity of OD being observed in the delta variant. Considering the fact that numerous further SARS-CoV-2 variants exist in the meantime, divergent data on the prevalence of OD may in fact partly be explained by a variable pathogenicity of these. Beyond that, there might also be possible ethnic differences in the pathogenicity of the different variants and a different host genetic susceptibility that contribute to the discrepancy of smell dysfunction, which are not represented in our study population [42,43]. Finally, olfactory function is also influenced by numerous nonviral baseline and clinical factors as indicated by our cross-sectional analysis, which in turn might also cause certain discrepancies regarding the prevalence of OD in different analyses.

In contrast to our results in COVID-19 negative patients, we did not observe a significant association between OD and age during a SARS-CoV-2 infection. In contrast, our longitudinal study results revealed a highly significant association between OD and the male gender as well as between OD and positive smoking status. Regarding association between OD and male gender, our results reflect what was already observed in our cross-sectional study cohort. However, in contrast to the relatively clear data situation in SARS-CoV-2 negative tested individuals, gender-specific loss of smell in SARS-CoV-2 infections is still controversial in the literature [42,44,45]. Causes for this discrepancy of test results may again be a test variability, as well as differences derived by different viral variants. Regarding the positive association of OD and smokers in SARS-CoV-2 infected patients, similar observations have already been made in the general population [46,47]. One possible theory for this association may be the fact that tobacco use induces squamous metaplasia or lesions of the olfactory mucosa, thereby predisposing to OD in general and in SARS-CoV-2 infections [47]. Beyond that, the positive association between OD and nicotine might be explained by the cholinergic pathway: nicotine is a cholinergic agonist and an important inhibitor of proinflammatory cytokines acting through the cholinergic anti-inflammatory pathway [48]. Involvement of the limbic cholinergic system in turn has been observed in olfactory processing in animal studies and in Alzheimer’s disease and thus may also play an important role in SARS-CoV-2 mediated OD [49]. A possible role of the parasympathetic pathway has also been observed in COVID-19 and some authors saw parallels in the clinical manifestation of toxicological mechanisms acting via the cholinergic pathway [50].

Our study has some limitations. First, our SARS-CoV-2 negative cohort predominantly consisted of patients aged 65 years and older, who had evidence of a major olfactory impairment. To limit this bias, we conducted sub-analyses of OD prevalence among different age groups. Second, all clinical parameters as well as their expression were recorded by questionnaires and not by medical examination. Due to the subjective nature of this survey, some of the parameters may be underestimated. Third, testing on SARS-CoV-2 was conducted by using RT-PCR-based diagnostic tests of naso- and oropharyngeal samples, which is considered as golden standard in diagnostic routine and has a high sensitivity of up to 63%. Nevertheless, a SARS-CoV-2 infection could not be ruled out with absolute certainty by using this test in close contacts.

## 5. Conclusions

To conclude, our findings demonstrate an OD prevalence of 45.7% in a large cohort of SARS-CoV-2 negative individuals with significantly higher rates in the older generation, male subjects and patients with CVD and pulmonary comorbidities. However, in a multivariate analysis, only age and gender turned out to be independent predictive factors of OD in COVID-19 negative tested patients. Among 90 SARS-CoV-2 positive tested individuals, OD was found in 65.6% of individuals with a significantly different prevalence and severity being observed between different SARS-CoV-2 variants. Male patients and patients with positive smoking status were observed to experience smell dysfunction significantly more frequently during SARS-CoV-2 infection. Due to the high variability of OD prevalence between different viral variants, the relatively low sensitivity and specificity of OD in predicting SARS-CoV-2 infections in close contacts of index cases and due to quite high prevalence of OD in the SARS-CoV-2 negative tested patients, olfactory testing might not be reliable enough to be implemented in diagnostic routine to early detect a SARS-CoV-2 infection. Moreover, our findings highlight the importance of considering different SARS-CoV-2 variants when analyzing and characterizing olfactory outcomes and suggest that OD might not be a hallmark of all SARS-CoV-2 variants.

## Figures and Tables

**Figure 1 jcm-12-03162-f001:**
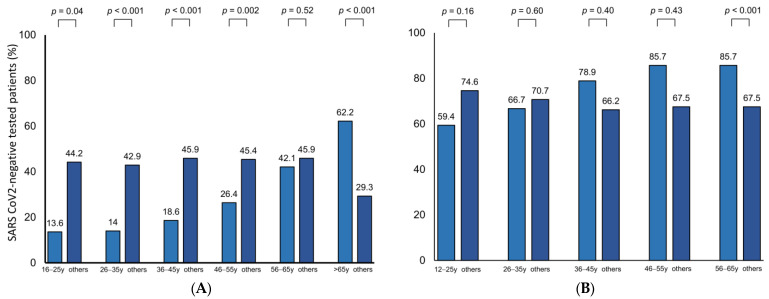
Prevalence of olfactory dysfunction according to different age groups.Prevalence of OD in (**A**) SARS CoV-2 negative tested individuals (*n* = 494) and in (**B**) SARS CoV-2 positive tested patients (*n* = 90).

**Table 1 jcm-12-03162-t001:** Clinical characteristics and comorbidities of COVID-negative patients (cross-sectional study part).

Characteristics	Patients (*n* = 524)
Patient age at diagnosis (years)	
16–25, *n* (%)	22 (4.2)
26–35, *n* (%)	43 (8.2)
36–45, *n* (%)	59 (11.3)
46–55, *n* (%)	72 (13.7)
56–65, *n* (%)	111 (21.2)
>65, *n* (%)	217 (41.4)
Gender	
Female, *n* (%)	204 (38.9)
Male, *n* (%)	320 (61.1)
Current smoking	
Yes, *n* (%)	92 (17.6)
No, *n* (%)	432 (82.4)
Comorbidities	
Diabetes, *n* (%)	101 (19)
Cardio-vascular disease, *n* (%)	237 (45)
Malignancy, *n* (%)	126 (24)
Asthma bronchiale, *n* (%)	10 (2)
Chronic obstructive pulmonary disease, *n* (%)	43 (8)
Chronic kidney failure, *n* (%)	69 (13)
Liver cirrhosis, *n* (%)	53 (10)
Immunodeficiency, *n* (%)	27 (5)
Inflammatory bowel disease (IBD), *n* (%)	10 (2)
Pre-operations in the ENT tract, *n* (%)	3 (0.6)
Severity of OD	
No OD (grade 0), *n* (%)	299 (57.1)
Mild OD (grade 1), *n* (%)	105 (20.0)
Moderate OD (grade 2), *n* (%)	65 (12.4)
Severe OD (grade 3), *n* (%)	34 (6.5)
Anosmia (grade 4), *n* (%)	21 (4.0)

COVID; coronavirus disease; ENT, ear, nose and throat; IBD, inflammatory bowel disease; OD, olfactory disease.

**Table 2 jcm-12-03162-t002:** Analysis of factors associated with olfactory dysfunction (*n* = 524).

	Olfactory Dysfunction (*n* = 226)	No Olfactory Dysfunction (*n* = 268)	*p* Value
Males, *n* (%)	158 (69.9)	149 (55.6)	<0.001
Age (years), median (range)	67 (16–94)	53 (21–86)	<0.001
Current smoking, *n* (%)	41 (18.1)	46 (17.2)	0.81
Comorbidities			
Diabetes mellitus, *n* (%)	52 (23.0)	49 (18.3)	0.22
Pulmonary diseases, *n* (%)	33 (14.6)	20 (7.5)	0.03
Cardiovascular diseases, *n* (%)	134 (59.3)	103 (38.4)	<0.001
Malignancy, *n* (%)	57 (24.9)	69 (25.7)	0.92
Renal diseases, *n* (%)	33 (14.6)	36 (13.4)	0.79
Hepatological diseases, *n* (%)	21 (9.3)	32 (12.7)	0.24
Preoperations in the ENT tract, *n* (%)	1 (0.4)	2 (0.7)	1.00

Abbreviations: ENT, ear nose throat.

**Table 3 jcm-12-03162-t003:** Clinical characteristics and general symptoms at the onset of the disease (longitudinal study part).

Clinical Characteristics and General Symptoms	Patients (*n* = 90)
Patient age at diagnosis (years)	
7–25, *n* (%)	32 (35.6)
26–35, *n* (%)	22 (24.4)
36–45, *n* (%)	21 (23.3)
46–55, *n* (%)	7 (7.8)
56–65, *n* (%)	8 (8.9)
Gender	
Female, *n* (%)	49 (54.4)
Male, *n* (%)	41 (45.6)
Current smoking	
Yes, *n* (%)	22 (24.4)
No, *n* (%)	73 (75.6)
General symptoms	
Fever, *n* (%)	9 (10)
Fatigue, *n* (%)	23 (25.6)
Myalgia or arthralgia, *n* (%)	12 (13.3)
Loss of appetite	1 (1.1)
ENT	
Sore throat	13 (14.4)
Nasal obstruction	50 (55.6)
Rinorrhea	50 (55.6)
Pneumological	
Cough	43 (47.8)
Shortness of breath	1 (1.1)
Digestive	
Nausea	1 (1.1)
Vomiting	1 (1.1)
Diarrhea	1 (1.1)
Neurological	
Smell disorders	7(7.8)
Taste disorders	3 (2.2)
Headache	27 (30)
Severity of clinical symptoms during course of the disease	
No symptoms, *n* (%)	20 (22.2)
Mild symptoms, *n* (%)	16 (17.8)
Moderate symptoms, *n* (%)	24 (26.7)
Severe Symptoms, *n* (%)	30 (33.3)

Abbreviations: ENT, ear nose throat.

**Table 4 jcm-12-03162-t004:** Analysis of factors associated with olfactory dysfunction (longitudinal study part) (*n* = 90).

Characteristics	Olfactory Dysfunction (*n* = 59)	No Olfactory Dysfunction (*n* = 31)	*p* Value
Males	31 (52.5)	7 (22.6)	0.04
Age (years), median (range)	32 (12–65)	26 (12–58)	0.08
Positive smoking history, *n* (%)	15 (25.4)	6 (19.4)	0.04
General symptoms			
Fever, *n* (%)	8 (13.6)	1 (3.2)	0.26
Fatigue, *n* (%)	18 (30.5)	5 (16.1)	0.31
Myalgia or arthralgia, *n* (%)	11 (18.6)	1 (3.2)	0.09
Loss of appetite, *n* (%)	0 (0)	1 (3.2)	0.41
ENT symptoms			
Sore throat, *n* (%)	12 (38.9)	1 (3.2)	0.06
Nasal obstruction, *n* (%)	33 (55.9)	13 (14.4)	0.26
Rhinorrhea, *n* (%)	33 (55.9)	13 (14.4)	0.65
Pneumological			
Cough, *n* (%)	33 (55.9)	10 (32.3)	0.17
Shortness of breath, *n* (%)	1 (1.7)	0 (0)	1.00
Digestive GI symptoms (nausea, vomiting, diarrhea), *n* (%)	0 (0)	1 (3.2)	0.31
Neurological			
Taste disorders, *n* (%)	2 (3.4)	1 (3.2)	1.00
Headache, *n* (%)	19 (32.2)	8 (25.8)	1.00

Abbreviations: ENT, ear nose throat; GI symptoms, gastrointestinal symptoms.

## Data Availability

The data are available upon request from the corresponding author.

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
