# Peer review of "Is Olfactory Testing a Useful Diagnostic Tool to Identify SARS-CoV-2 Infections Early? A Cross-Sectional and Longitudinal Analysis"

_jcm, 2023, doi:10.3390/jcm12093162_

Round 1

Reviewer 1 Report

Authors Graf C. et al. in the present manuscript entitled “ Is olfactory testing a useful diagnostic tool to early identify SARS CoV-2 infections: a cross-sectional and longitudinal analysis”,  highlight, through an observational but also longitudinal study, the interactions of the dysfunctions of the sense of smell in the study sample population during the pandemic of COVID-19. Their aim is to test whether the loss-of-smell screening test can be useful as a diagnostic test of COVID-19. The work is well structured except for a few things to be improved. With this manuscript, the authors can make a difference from many other studies published so far on the same topic. I strongly recommend that they should strive to make it excellent. It is an interesting study and deserves to be published but only after the authors have resolved the few major revisions and minor revisions that I pointed out.

 Major revisions:

  1. Line 45, please add all symptoms of COVID-19, including also neurological ones, and cite 2,3 references.
    •  Https://www.covid19treatmentguidelines.nih.gov/overview/clinical-spectrum/#:~:text=Mild%20illness%3A%20Individuals%20who%20have,dyspnea%2C%20or%20abnormal%20chest%20imaging.
    • Baig AM, Greig NH, Gerlach J, Salunke P, Fabrowski M, Viduto V, Ali T. Underlying Causes and Treatment Modalities for Neurological Deficits in COVID-19 and Long-COVID. ACS Chem Neurosci. 2022 Oct 3. doi: 10.1021/acschemneuro.2c00482. Epub ahead of print. PMID: 36190929.

2.      Please, in addition to reference 4, fully describe what the pre-pandemic scientific literature also says about the sense of smell and the medical conditions under which it can be observed. The authors could use the following articles:

·        Graziadei PP, Karlan MS, Graziadei GA, Bernstein JJ. Neurogenesis of sensory neurons in the primate olfactory system after section of the fila olfactoria. Brain Res. 1980 Mar 31;186(2):289-300. doi: 10.1016/0006-8993(80)90976-2. PMID: 6766784

·        Baba T. et al., «Severe olfactory dysfunction is a prodromal symptom of dementia associated with Parkinson’s disease: a 3-year longitudinal study», Brain J. Neurol., vol. 135, n. Pt 1, pp 161–169, gen. 2012, doi: 10.1093/brain/awr321.

·        Silveira-Moriyama L. et al., Regional differences in the severity of Lewy body pathology across the olfactory cortex, Neurosci. Lett., vol. 453, n. 2, page. 77–80, apr. 2009, doi: 10.1016/j.neulet.2009.02.006.

·        Hubbard P. S., Esiri M. M., Reading M., McShane R.,. Nagy Z, Alpha-synuclein pathology in the olfactory pathways of dementia patients, J. Anat., vol. 211, n. 1, pp. 117–124, 2007, doi: 10.1111/j.1469-7580.2007.00748.x

·        Bucaretchi F, Borrasca-Fernandes CF, De Capitani EM, Hyslop S. Consecutive envenomation of two men bitten by the same coral snake (Micrurus corallinus). Clin Toxicol (Phila). 2020 Feb;58(2):132-135. doi: 10.1080/15563650.2019.1610568. Epub 2019 May 13. PMID: 31079507.

·        Sethi M, Cook M, Winkel KD. Persistent anosmia and olfactory bulb atrophy after mulga (Pseudechis australis) snakebite. J Clin Neurosci. 2016 Jul;29:199-201. doi: 10.1016/j.jocn.2015.12.019. Epub 2016 Feb 17. PMID: 26896910.

·        Gutiérrez JM, Calvete JJ, Habib AG, Harrison RA, Williams DJ, Warrell DA. Snakebite envenoming. Nat Rev Dis Primers. 2017 Sep 14;3:17063. doi: 10.1038/nrdp.2017.63. Erratum in: Nat Rev Dis Primers. 2017 Oct 05;3:17079. PMID: 28905944.

·        Pearn J, McGuire B, McGuire L, Richardson P. The envenomation syndrome caused by the Australian Red-bellied Black Snake Pseudechis porphyriacus. Toxicon. 2000 Dec; 38(12):1715-29. doi: 10.1016/s0041-0101(00)00102-1. PMID: 10858512

·        Upadhyay U. D. , Holbrook E. H., Olfactory loss as a result of toxic exposure, Otolaryngol. Clin. North Am., vol. 37, n. 6, pp. 1185–1207, dic. 2004, doi: 10.1016/j.otc.2004.05.003.

Hansen D. Anosmie nach Grippe [Anosmia following influenza]. Munch Med Wochenschr. 1970;112(48):2167-2169.

Akerlund A, Bende M, Murphy C. Olfactory threshold and nasal mucosal changes in experimentally induced common cold. Acta Otolaryngol. 1995;115(1):88-92. doi:10.3109/00016489509133353

There are other past studies highlighting Anosmia with other viruses. It is important to consider this as well.  

  1. Line 56-58: what is stated is partially true, and a reference should be added. It should be considered that many studies indicate fecal molecular testing is also more sensitive.

a.      Mardian Y., Kosasih H., Karyana M., Neal A., Lau C.-Y., «Review of Current COVID-19 Diagnostics and Opportunities for Further Development”, Front. Med., vol. 8, 2021, doi: 10.3389/fmed.2021.615099.

b.      Natarajan, A. et al. “Gastrointestinal symptoms and fecal shedding of SARS-CoV-2 RNA suggest prolonged gastrointestinal infection.” Med (New York, N.Y.) vol. 3,6 (2022): 371-387.e9. doi:10.1016/j.medj.2022.04.001

c.      Petrillo M., Brogna C., Cristoni S., Querci M., Piazza O., G. Van den Eede, “Increase of SARS-CoV-2 RNA load in faecal samples prompts for rethinking of SARS-CoV-2 biology and COVID-19 epidemiology”, F1000Res, vol. 10, pp. 370,  2021, doi: 10.12688/f1000research.52540.1.

Discuss well both nasal and fecal tests and any limitations.

  1. Extend the introduction more by explaining both airborne and oral-fecal transmission routes

5.      Line 156-188 and figure s1 A show the percentage of subjects with no, mild, moderate, and severe symptoms. In that case, the authors should cite well the difference on how they classified patients by symptom severity and report references. The WHO w NIH have slight differences in classifying moderate or mild patients. Please make attention to these suggested references as well.

  1. Line 109-111. How do they confidently rule out the non-presence of the virus if the family members tested negative on the nasal test? Please, the authors need to correct this and remember that SARS-CoV-2 is found in both feces and wastewater, and many studies find it in the feces of patients for up to 30 days.
  2. Line 197 and line 200: when the authors use the term Delta variant and Omicron variant, is it because they sequenced the viral genome in each patient, or is it because the patients became ill during the period when these variants were circulating? Please specify. If they sequenced the samples with RNA virus, it would be important to see the data; otherwise, it should be specified that the patients were observed when one type of variant was circulating more than another.
  3. Line 202-203: This is a very interesting point. Although many studies hypothesize that SARS-CoV-2 replicates in olfactory bulb cells (add references), and although the current authors find divergence between the Delta and Omicron variants, it would be important for them to emphasize this point. In some cases molecular nasal, oro-pharyngeal, and saliva swabs tests are positive, and not everyone has the loss of the sense of smell moreover, not everyone recovers at the same time when this sense is disrupted (4 cases are not shown in the table because they recovered beyond 28 days, it seems to be my understanding). These data allow us to hypothesize other mechanisms on losing the sense of smell that is not yet fully understood. It would be helpful to the scientific community if the authors would give a nod about it
  4. Line 235-240: The authors should also argue the neurological disorders associated with loss of taste and smell, such as Parkinson's, Alzheimer's, and other central neuronal type diseases.
  5. Line 290-292: this point is crucial, and the authors can add something considerable.  It is necessary to think that Studies conducted over the past decade have established that regeneration of the olfactory bulb takes 60 to 90 days in the animal model (Graziadei PP, Karlan MS, Graziadei GA, Bernstein JJ. Neurogenesis of sensory neurons in the primate olfactory system after section of the fila olfactoria. Brain Res. 1980 Mar 31;186(2):289-300. doi: 10.1016/0006-8993(80)90976-2. PMID: 6766784) and Hyposmia is a symptom that appears in many other special conditions, such as in some autoimmune diseases (PD, AZ, schizophrenia and others), in exposure to toxins or toxic substances, and others.
  6. Line 311: At this point, the authors should also analyze other mechanisms  (toxicologic aspect) as recently published: Brogna C, Cristoni S, Brogna B, Bisaccia DR, Marino G, Viduto V, Montano L, Piscopo M. Toxin-like Peptides from the Bacterial Cultures Derived from Gut Microbiome Infected by SARS-CoV-2—New Data for a Possible Role in the Long COVID Pattern. Biomedicines. 2023; 11(1):87. https://doi.org/10.3390/biomedicines11010087

  1. Line 312: it is also important to consider the cholinergic pathway suggested by Farsalinos, Konstantinos et al. “Editorial: Nicotine and SARS-CoV-2: COVID-19 may be a disease of the nicotinic cholinergic system.” Toxicology reports vol. 7 658-663. 30 Apr. 2020, doi:10.1016/j.toxrep.2020.04.012

Minor revision

1.      Probably a question mark is missing in the title.

2.      Please add the lineage (  https://cov-lineages.org/) near to the name of the variants of SARS-CoV-2: line 28

3.      Please add the geographic location where the study was conducted: lines 17-18 “ during SARS-CoV-2 infection in ……..”

4.      In the abstract, add hyphens between the words SARS-CoV-2 and in all the manuscript where it is missing.

5.      Add to reference 3 a few clinical study references, line 47

6.      Line 47, 48, please add a reference.

7.      Line 52-54: Remodel the sentence and add a sentence.

8.      Line 55. Add  references

9.      Line 64, Describe some olfactory tests routinely practiced in the medical clinic and cite some references.

10.   u-Smell-it, line 67, please add reference number 5

11.   Line 145,150 CVD:  replace or add cardiovascular disease

12.   Line 197, please write RKI in its entirety

Author Response

Dear editors,

first, we would like to thank you and the referees for the assessment and consideration of our work, as well as the opportunity to revise our manuscript. The reviewers raised a number of important questions, which we addressed in the revised manuscript and in the point-by-point letter. The wording was changed in order to improve the quality and intelligibility of our manuscript. Please see our point-by-point answers enclosed.

Once again we would like to thank you and the reviewers for contributing to the significant improvement of our script and we kindly hope that it now meets the criteria for publication in Journal of Clinical Medicine.    

We would be grateful for consideration of our manuscript for publication in Journal of Clinical Medicine and look forward to hearing from you.

Yours sincerely

Sandra Ciesek and Christiana Graf

Reviewer 2 Report

1.   More correct according to statistics recommendations if the control group and the main group consisted of approximately the same number of subjects. Half of the participants should with SARS CoV-2 negative tested patients, while the other half is SARS CoV-2 positive tested patients, as this strategy gets us equally-precise estimates of the mean or frequency of the dependent variable in both conditions.

2.     Inclusion in the study  groups of different sizes: a group of patients with a negative test for SARS CoV-2 84.6% of patients and a group of patients with a positive test for SARS CoV-2 15.45% of patients, would make the standard error higher in the second group and make it difficult to determine if there are significant differences between the 2 groups.

3.     In addition, the use of different research methods in the study of each group (sectional-cross and longitudinal study) may not reflect the objective differences between the two groups. Each group must have equal conditions: contingent location and  dates of the event. In the text was indicated that the cross-sectional study part was conducted in the University Hospital of Frankfurt, but the longitudinal study part was investigated on an outpatient basis by a collaboration with the Frankfurt Health Department.

4.     Smoking is a risk factor for OD. Perhaps it would be better if you excluded these patients from the first group (part of the cross-sectional study)? Or compare the results of this part with the results of a longitudinal study without these patients.

5.     How do you explain the large number of patients with OD in the first study (section-cross study)?

6.     In sectional-cross study most of the observation is over 65 years of age. These patients often show signs of polyneuropathy with the development of OD. Moreover, in the second study (Longitudinal study) there is not a single patient over 65 years old!!

Thsnks!

Author Response

(The authors gave the same response as above.)

Reviewer 3 Report

The paper is interesting and deserves attention. It points out a new insight on the topic of olfactory involvement in COVID-19-associated syndrome, retrieving results that are somewhat in contrast with the scientific literature to date and with the common behalf. Some of the main features of the present paper are meritory, making it a useful piece of science to be considered. However, frankly speaking, there are a number of points to be highlighted in the evaluation of the present work, as follows:

- The Introduction should better go into depth of the pathophysiology of Sars-CoV-2 syndrome and its links, justifying a possible association with the olfactory involvement.

- Methodology is sub-optimal. Indeed, despite being aware of the many advantages of the u-Smell-it Test (e.g., easy, fast administration, etc.), the test is affected by a number of intrinsical methodological issues, making it ideal for screening purposes, but not to generalize its findings. This is because of, for example, the low number of items composing the test, which can represent the ideal milieu for ceiling effects or vice-versa, which are much less present in other tests, like the Sniffin' Sticks or, even more, the UPSIT. Furthermore, it is noticed by the authors in the limitations that the population is composed by a good number of "older" people, making the comparison more tricky, as the negative individuals are often experiencing an olfactory deficit, too, associated with their age-related decline. This is an important limitation of the methodology, notably of the sample study selection.

Therefore, I would recommend a major revision to allow authors improving the points mentioned and adding more justification, when the limitation is present, or the limitation itself, when actually not present, to the paper.

Author Response

dear editors,

first, we would like to thank you and the referees for the assessment and consideration of our work, as well as the opportunity to revise our manuscript. The reviewers raised a number of important questions, which we addressed in the revised manuscript and in the point-by-point letter. The wording was changed in order to improve the quality and intelligibility of our manuscript. Please see our point-by-point answers enclosed.

Once again we would like to thank you and the reviewers for contributing to the significant improvement of our script and we kindly hope that it now meets the criteria for publication in Journal of Clinical Medicine.    

We would be grateful for consideration of our manuscript for publication in Journal of Clinical Medicine and look forward to hearing from you.

Yours sincerely

Sandra Ciesek and Christiana Graf

Round 2

Reviewer 1 Report

The authors have improved the manuscript.

Author Response

Dear editors,

first, we would like to thank you and the referees for the assessment and consideration of our work, as well as the opportunity to revise our manuscript. The reviewer raised a number of important questions, which we addressed in the revised manuscript and in the point-by-point letter. The wording was changed in order to improve the quality and intelligibility of our manuscript. Please see our point-by-point answers enclosed.

Once again we would like to thank you and the reviewers for contributing to the significant improvement of our script and we kindly hope that it now meets the criteria for publication in Journal of Clinical Medicine.    

We would be grateful for consideration of our manuscript for publication in Journal of Clinical Medicine and look forward to hearing from you.

Yours sincerely

Sandra Ciesek and Christiana Graf

Reviewer 2 Report

“Is olfactory testing a useful diagnostic tool to early identify SARS-CoV-2 infections:”  is the main question addressed by the research and consider the topic original and  relevant in the field.

The manuscript is clear and relevant to the field and is presented in a well-structured manner. The manuscript has a scientific basis.

Fiigures, tables, images and schemes are appropriate, they properly show the data and they easy to interpret and understand. 

All my statistical questions were answered in detail. And answers were quite convincing.

Additional materials and explanations in the abstract, introduction, materials and methods, results and discussion made the article better from a scientific point of view with more convincing argument

There is no conclusion. Add please, according to the title and purpose of your work.  

Figures under the numbers 1 are not presented according to the rules. The letters (a) (b) must be in bold and below the figures. Better look at the MDPI. log rules.

Add this caption “SARS CoV2-positive tested patients (%)” to the vertical axis of the image (b).

References are not formed by the MDPI recommendation. There must be a dot between the journal title and the year of publication. The title of the journal should be in italics and the year in bold.

Author Response

(The authors gave the same response as above.)

Reviewer 3 Report

The authors successfully addressed my concerns.

Author Response

(The authors gave the same response as above.)
